# 4-Phenylbutyric Acid Improves Gait Ability of UBAP1-Related Spastic Paraplegia Mouse Model: Therapeutic Potential for SPG80

**DOI:** 10.3390/ijms26199779

**Published:** 2025-10-08

**Authors:** Keisuke Shimozono, Yeon-Jeong Kim, Takanori Hata, Haitian Nan, Kozo Saito, Yasunori Mori, Yuji Ueno, Fujio Isono, Masaru Iwasaki, Schuichi Koizumi, Toshihisa Ohtsuka, Yoshihisa Takiyama

**Affiliations:** 1Department of Neurology, Graduate School of Medical Sciences, University of Yamanashi, Yamanashi 409-3898, Japan; kshimozono@yamanashi.ac.jp (K.S.); thata@yamanashi.ac.jp (T.H.); uenoy@yamanashi.ac.jp (Y.U.); 2Department of Biochemistry, Graduate School of Medical Sciences, University of Yamanashi, Yamanashi 409-3898, Japan; kimuy@yamanashi.ac.jp (Y.-J.K.); yasmori@yamanashi.ac.jp (Y.M.); tohtsuka@yamanashi.ac.jp (T.O.); 3Department of Neurology, Xuanwu Hospital, Capital Medical University, 45 Changchun Street, Beijing 100053, China; poseidon_1987427@sina.com; 4Department of Neuropharmacology, Graduate School of Medical Sciences, University of Yamanashi, Yamanashi 409-3898, Japan; ksaitoko@yamanashi.ac.jp (K.S.); skoizumi@yamanashi.ac.jp (S.K.); 5Centre for Advancing Clinical Research (CACR), Graduate School of Medical Sciences, University of Yamanashi, Yamanashi 409-3898, Japan; fisono@yamanashi.ac.jp; 6Organization for the Promotion of Research and Social Collaboration, University of Yamanashi, Yamanashi 409-3898, Japan; miwasaki@yamanashi.ac.jp; 7Department of Neurology, Fuefuki Central Hospital, Yamanashi 406-0032, Japan

**Keywords:** hereditary spastic paraplegia (HSP), SPG80, *UBAP1*, 4-phenylbutyric acid (4-PBA), ER stress, microglia activation

## Abstract

Spastic paraplegia 80 (SPG80), caused by mutations in *ubiquitin-associated protein 1 (UBAP1)*, is a pure form of juvenile-onset hereditary spastic paraplegia (HSP) and leads to progressive motor dysfunction. Despite recent advances in the molecular analyses of HSP, disease-modifying therapy has not been established for HSP including SPG80. In the present study, we evaluated the therapeutic potential of 4-phenylbutyric acid (4-PBA), a chemical chaperone and histone deacetylase inhibitor, in *Ubap1* knock-in (KI) mice expressing a disease-associated truncated UBAP1 variant. We found that 4-PBA administration significantly improved the motor performance of KI mice in the rotarod and beam walk tests, with maximal benefits achieved when given during pre- or early-symptomatic stages. Partial efficacy was also observed when treatment began after symptom onset in KI mice. Furthermore, 4-PBA attenuated spinal microglial activation and partially restored microglial morphology, although astrocytic reactivity remained unchanged. These findings support 4-PBA as a candidate therapeutic compound for SPG80 and highlight the potential of proteostasis-targeted interventions in HSPs.

## 1. Introduction

Hereditary spastic paraplegia (HSP) is a clinically and genetically diverse group of neurodegenerative disorders characterized by progressive spasticity and weakness of the lower extremities [1]. Clinically, HSP is classified into a pure form, which is characterized by leg spasticity accompanied by bladder disturbance and disturbance of the vibration sense, and a complicated form, which is characterized by leg spasticity and many complications such as ataxia, a thin corpus callosum, chorioretinal dystrophy, intellectual disability, extrapyramidal signs, and peripheral neuropathy [1,2]. These disorders primarily result from the degeneration of corticospinal tracts, with over 90 associated genes or loci identified to date [1,2,3]. At the molecular level, many HSP-associated genes are implicated in essential cellular processes, including intracellular transport, organelle morphology, mitochondrial dynamics, and proteostasis, particularly affecting long motor axons [4].

In 2019, we and other investigators independently identified a juvenile-onset, pure form of autosomal dominant HSP caused by mutations in *ubiquitin-associated protein 1 (UBAP1)* (SPG80) [5,6,7]. UBAP1 is a core component of the endosomal sorting complex required for transport-I (ESCRT-1), which recognizes ubiquitinated cargos in early endosomes and facilitates their incorporation into multivesicular bodies (MVBs) for lysosomal degradation [8,9]. Dysfunction of UBAP1 or other ESCRT components impairs MVB formation and endolysosomal trafficking, resulting in the accumulation of misfolded or ubiquitinated proteins, a hallmark of many neurodegenerative diseases [10,11,12]. In SPG80, impaired UBAP1 function likely disrupts endosomal maturation and protein clearance, thereby contributing to axonal degeneration [13].

A very recent study of a murine model of SPG15, a complicated form of HSP, has shown that microglial activation precedes neuronal loss, implicating neuroinflammation as a contributing factor in disease progression [14]. This finding suggests that glial activation, alongside impaired proteostasis, may also play a role in the pathogenesis of HSP, including SPG80. Given that intracellular stress and protein misfolding are known to trigger glial responses [15,16], therapeutic strategies aimed at restoring proteostasis may help mitigate neuroinflammation and neuronal damage in HSP [17].

Chemical chaperones that restore intracellular proteostasis are attracting attention as potential therapeutics for neurodegenerative diseases [18,19]. Among them, 4-phenylbutyric acid (4-PBA), a low-molecular-weight fatty acid analog initially developed to treat urea cycle disorders [20], has shown promise. 4-PBA efficiently crosses the blood–brain barrier, mitigates endoplasmic reticulum (ER) stress, and promotes protein folding and degradation of misfolded proteins, thereby exerting cytoprotective effects [19,20,21]. In addition, its histone deacetylase (HDAC) inhibitory activity may affect gene expression and enhance cellular stress responses [22].

We previously demonstrated that a *Ubap1^+/E176Efx23^* knock-in (KI) mouse model exhibits progressive gait impairment due to upper motor neuron dysfunction, with no other detectable physical abnormalities. These features closely replicate human SPG80, validating the usefulness of the model for mechanistic and therapeutic studies [13]. In this paper, we evaluated the therapeutic efficacy of 4-PBA in our KI model. By varying the treatment dosage and timing, we evaluated the effects of 4-PBA on motor performance and glial pathology. The present findings offer novel insights into the potential use of chemical chaperones as a treatment strategy for SPG80.

## 2. Results

### 2.1. Therapeutic Effects of 4-PBA on Motor Function and Coordination in Ubap1 KI Mice

Motor performance was evaluated using the accelerating rotarod test (Figure 1A and Appendix A). Untreated KI mice exhibited significantly shorter latency to fall than wild-type (WT) controls, indicating motor deficits. Among the treatment groups, mice receiving 1000 mg/kg/day from 2.5 to 4.5 months showed the greatest improvement (*p* < 0.05). Importantly, even when treatment was initiated at 4.5 months (after the typical onset of symptoms), partial functional recovery was observed. Our previous work indicated that most KI mice develop overt gait abnormalities by 4 months of age [13]. Thus, 4-PBA still exerts therapeutic potential even when administered after the onset of symptoms, highlighting its clinical relevance and flexible treatment windows.

To further evaluate motor coordination, we conducted the beam walking test using the most effective regimen (1000 mg/kg/day from 2.5 to 4.5 months). Untreated KI mice exhibited significantly more foot slips across three trials than WT mice, indicating deficits in motor coordination (Figure 1B and Appendix A). Notably however, 4-PBA treatment significantly reduced the number of foot slips (*p* < 0.05), confirming an improvement in coordination.

Longitudinal monitoring of body weight revealed no significant differences among WT, untreated KI, and 4-PBA-treated KI mice (Figure 1C). Additionally, no observable behavioral abnormalities or weight fluctuations were noted, indicating good tolerability to 4-PBA under the test conditions.

### 2.2. Glial Activation and Effects of 4-PBA on the Spinal Cord of Ubap1 KI Mice

To assess glial activation and the therapeutic effects of 4-PBA, we performed double immunofluorescence staining for ionized calcium binding adapter protein 1 (Iba1) and glial fibrillary acidic protein (GFAP)—established markers of microglia and astrocytes, respectively—in the spinal cords of KI mice (Figure 2A). This approach enabled us to evaluate cell-type-specific neuroinflammatory responses in the principal site of motor dysfunction and to determine whether 4-PBA modulates glial reactivity in a region directly implicated in SPG80 pathology. Compared with WT controls, KI mice showed marked upregulation of both Iba1 and GFAP. We focused our analysis on the anterior horn and white matter—spinal regions essential for motor control (Figure 2B,E). Quantitative analysis revealed significant increases in Iba1 signal intensity in both regions, which were partially attenuated by 4-PBA treatment (Figure 2B,D). In contrast, the GFAP expression level remained elevated and was not significantly affected by 4-PBA (Figure 2C,E).

Morphological analysis further confirmed microglial activation in KI mice, as evidenced by enlarged soma and reduced process complexity. These changes were significantly ameliorated by 4-PBA: soma size was reduced (Figure 2G), and Sholl analysis demonstrated increased branching complexity (Figure 2H), indicating a partial restoration of microglial homeostasis.

### 2.3. Microglial Morphology and Therapeutic Effects of 4-PBA on the Cerebral Cortex

To determine whether the glial alterations observed in the spinal cord extended to other regions of the central nervous system, we next examined microglial morphology and the effects of 4-PBA on the cerebral cortex. Immunofluorescence staining for Iba1 and GFAP revealed no significant differences in the number or distribution of Iba1-positive microglia or GFAP-positive astrocytes between KI mice and wild-type (WT) controls (Figure 3A). Quantitative analysis confirmed that neither the density of Iba1-positive cells nor their regional coverage differed significantly between groups (Figure 3B,C), suggesting that microglial proliferation or recruitment is not markedly altered in the cortex.

Despite this, cortical microglia in KI mice exhibited distinct morphological changes, including somatic hypertrophy and reduced branching complexity (Figure 3D). Sholl analysis demonstrated a significant reduction in microglial arborization, which was partially reversed by 4-PBA treatment (Figure 3E). Similarly, 4-PBA tended to reduce soma size, although this effect did not reach statistical significance (*p* = 0.06) (Figure 3F). These findings indicate that microglial activation in the cortex occurs primarily at the morphological level rather than through increased cell number, and that 4-PBA can partially ameliorate these structural alterations.

## 3. Discussion

The present study demonstrates that 4-PBA improves motor function in the *Ubap1* KI mouse model, particularly when treatment begins at pre- or early-symptomatic stages. Even when administered after symptom onset, 4-PBA conferred measurable benefits. This finding is especially relevant to clinical settings, where diagnosis typically occurs post symptomatically. These findings support the hypothesis that impaired proteostasis accompanied by intracellular stress contributes to the progression of SPG80, and that 4-PBA acts by mitigating ER stress and promoting protein homeostasis [23,24]. The observation that motor deficits can be reversed, even partially, and post-symptomatically, highlights the potential of 4-PBA as a therapeutic candidate for human HSP.

In the spinal cord of KI mice, we observed microglial activation characterized by increased Iba1 expression levels, somatic hypertrophy, and reduced process complexity. These alterations were partially reversed by 4-PBA, while astrocyte activation—as indicated by the unchanged GFAP expression level—remained unaffected. Notably, although 4-PBA did not reduce the total number of Iba1-positive microglia, it partially restored their morphology, as evidenced by smaller cell bodies and more ramified processes. These findings suggest that 4-PBA attenuates microglia activation at the functional level, even though it did not normalize microglial numbers. Therefore, the improvement in motor performance may be attributed, at least in part, to modulation of microglial activity rather than to a complete rescue of microgliosis. This suggests a selective effect of 4-PBA on microglial regulation, likely through the reduction in proteotoxic stress and downstream inflammatory signaling [25,26].

The discrepancy between the robust behavioral improvements and the modest glial modulation suggests that 4-phenylbutyric acid (4-PBA) may exert additional protective effects on motoneurons. Although direct evidence for 4-PBA-mediated neuroprotection in motoneurons is currently limited, its established role in reducing endoplasmic reticulum stress and enhancing protein homeostasis provides a plausible mechanism by which it could support motoneuronal survival and function [27]. This may help explain the behavioral improvements observed in our study. To clarify the contribution of neuronal protection to the therapeutic effects of 4-PBA, future investigations should directly assess motoneuronal integrity and function using approaches such as immunostaining for choline acetyltransferase (ChAT) and NeuN, as well as electrophysiological recordings.

The differences in responses between microglia and astrocytes may reflect distinct activation mechanisms: microglia respond to intracellular proteotoxic stress, whereas astrocyte reactivity is more affected by extracellular signals and cytokines [28,29]. These observations align with a previous study demonstrating that 4-PBA enhances neuronal function but does not fully mitigate astrogliosis [27]. Consequently, targeting astrocyte activation may necessitate complementary strategies, such as anti-inflammatory or cytokine-directed therapies.

In contrast to the spinal cord, cortical glial responses were relatively mild. Despite morphological alterations in cortical microglia, there was no significant increase in glial marker expression level, consistent with milder cortical pathology in other HSP models such as SPG11 and SPG15 [30]. This regional specificity reflects the clinical phenotype of SPG80, which primarily affects lower limb function mediated by spinal motor pathways.

Our prior work showed that *UBAP1* mutations impair endosomal trafficking, leading to the accumulation of ubiquitinated proteins and axonal degeneration [13]. In the present study, 4-PBA treatment improved motor function and attenuated microglial activation in UBAP1-deficient mice, suggesting a beneficial effect on the spinal neuroimmune environment. These observations are consistent with a potential enhancement of proteostasis, although direct evidence for reduced ubiquitinated protein accumulation was not assessed. Given that activated microglia exacerbate neuronal stress and degeneration [14,31,32], the ability of 4-PBA to modulate microglial activity may be an important component of its therapeutic action.

While our study focused on UBAP1-related HSP, the beneficial effects of 4-PBA on proteostasis and motor function may also be relevant to other HSP subtypes, such as SPG11, SPG15, SPG4, SPG20, and SPG53, which involve microglial activation, lysosomal stress, or ESCRT dysfunction [14,33,34,35,36,37]. Given that these pathological features converge on impaired proteostasis, chemical chaperones like 4-PBA could offer broader therapeutic potential. However, direct experimental validation in each subtype will be necessary to confirm this possibility.

In summary, 4-PBA alleviates motor deficits and spinal microgliosis in a *UBAP1* mutant model of SPG80. Although it does not completely reverse the underlying cellular pathology, the observed improvements in motor performance and glial activation highlight its therapeutic potential. These findings support further investigation into the molecular basis of UBAP1-related diseases and the development of multimodal strategies, including gene therapy and proteostasis-targeting combination treatments.

Importantly, although 4-PBA significantly improved motor function in *Ubap1* mutant mice, caution is warranted when extrapolating these findings to humans. 4-PBA is already approved for clinical use in urea cycle disorders [38] and has also been considered for neurodegenerative diseases [39,40,41], supporting its potential repurposing for HSP. However, several key issues remain unresolved, including optimal dosing, CNS penetration, and long-term safety in patients. Further studies would be required to investigate dose–response relationships and assess potential toxicity to ensure safe clinical translation. Thus, while our results provide preclinical evidence supporting 4-PBA as a candidate therapy for HSP, well-designed clinical trials would ultimately be necessary to establish its therapeutic efficacy in patients.

## 4. Materials and Methods

### 4.1. Animal Experiments

All experiments were performed using C57BL/6J mice obtained from SLC (Hamamatsu, Japan). *Ubap1* KI mice were generated using the CRISPR/Cas9 system as previously reported [13]. Genotyping was performed by polymerase chain reaction using tail DNA. Mice were housed under standard conditions (24 ± 1 °C, 12 h light/dark cycle, lights on from 8 a.m. to 8 p.m.) with ad libitum food and water. Only male mice were used to avoid estrous cycle variability. WT littermates from heterozygous crosses served as controls.

### 4.2. Drug Administration Timeline

To assess the therapeutic efficacy of 4-PBA, we employed four distinct treatment regimens in KI mice, varying in both dosage and duration. Mice received oral 4-PBA (sodium 4-phenylbutyrate; Cat. No. 11323, Cayman Chemical, Ann Arbor, MI, USA) at either 500 mg/kg/day or 1000 mg/kg/day, beginning at 2.5 or 4.5 months of age. Specifically, the regimens were as follows: 500 mg/kg/day for 1 month starting at 2.5 months; 1000 mg/kg/day for 1 month starting at 2.5 months; 1000 mg/kg/day for 2 months starting at 2.5 months; and 1000 mg/kg/day for 2 months starting at 4.5 months (Figure 4). These treatment windows were chosen to evaluate both early intervention and delayed administration after the emergence of motor symptoms. The doses were selected on the basis of previous studies demonstrating efficacy and tolerability in mouse models [19,21,27,42].

### 4.3. Behavioral Tests

Motor function was assessed by rotarod and beam walk tests. In the beam walk test, 2- to 6-month-old male mice traversed a 1-cm diameter, 100-cm long stainless-steel bar elevated 70 cm above the ground. Foot slips were recorded over three trials, with a maximum score of 10 per trial.

In rotarod tests, mice were evaluated using an accelerated rotarod protocol (Muromachi, MK-610A, Tokyo, Japan). The rotation speed increased linearly from 4 to 40 rpm over 300 s, and the average latency to fall was calculated from three consecutive trials. Mice were trained prior to the test sessions. We employed the accelerated paradigm because it provides higher sensitivity to detect subtle motor impairments in HSP models compared with constant-speed protocols, as reported in the previous studies [43,44].

### 4.4. Tissue Preparation and Immunofluorescence Staining

Mice were anesthetized by the intraperitoneal injection of medetomidine, midazolam, and butorphanol, followed by transcardial perfusion with ice-cold phosphate-buffered saline (PBS, 0.1 M, pH 7.4) and 4% paraformaldehyde (PFA) in PBS. Brains and spinal cords were post fixed in 4% PFA overnight at 4 °C, and then cryoprotected in a 30% *w*/*v* sucrose solution for two days. Tissues were sectioned at 20 μm thickness using a cryostat (CM1520, Leica Biosystems, Nussloch, Germany). Sections were washed three times for 5 min each in PBS, blocked with 0.1% Triton X-100/10% NGS for 1 h at room temperature, and then incubated with primary antibodies in PBS at 4 °C for 24 h.

The primary antibodies used were rabbit anti-Iba1 (1:500; WAKO, Cat# 019-19741, Osaka, Japan) and mouse anti-GFAP (1:500; WAKO, Cat# MO389, Osaka, Japan). After the conjugation of the tissues with the primary antibodies, the sections were washed three times for 5 min each in PBS and then incubated with the corresponding secondary antibodies for 2 h at room temperature. The secondary antibodies were Alexa Fluor 546-conjugated goat anti-rabbit IgG (H+L) (1:1000; Thermo Fisher Scientific, Cat# A10035, Waltham, MA, USA) and Alexa Fluor 488-conjugated goat anti-mouse IgG (H+L) (1:1000; Thermo Fisher Scientific, Cat# A28175, Waltham, MA, USA).

For Sholl analysis and soma size measurements, microglial cells were selected from randomly chosen fields within the anterior horn of the spinal cord and the cerebral cortex. To minimize selection bias, only cells with clearly distinguishable and intact somata and processes, without overlap with neighboring cells, were included in the analysis.

### 4.5. Sholl Analysis

Dendritic complexity was quantified using the Sholl Analysis plugin in ImageJ (version 1.54p, NIH, Bethesda, MD, USA). Neurons were traced using the Simple Neurite Tracer plugin, with centric circles drawn at 10 μm intervals from the soma. The number of intersections at each radius was used to assess arbor complexity.

### 4.6. Statistical Analyses

Statistical analyses were performed using EZR (version 1.54), a graphical interface for R. Data normality was assessed by the Shapiro–Wilk test. ANOVA with Tukey’s or Dunnett’s post hoc tests was applied to normally distributed data. Non-parametric data were analyzed using Kruskal–Wallis followed by Bonferroni-corrected Mann–Whitney U tests. For Sholl analysis, group comparisons at each radial distance were conducted using Kruskal–Wallis tests followed by Bonferroni-corrected pairwise Mann–Whitney *U* tests.

## 5. Conclusions

Our study showed that 4-PBA partially restores motor function and attenuates microglial activation in KI mice with SPG80 expressing a truncated UBAP1 protein. Although not completely curative, 4-PBA shows promise as a component of future therapeutic strategies targeting proteostasis in HSP.

## Figures and Tables

**Figure 1 ijms-26-09779-f001:**
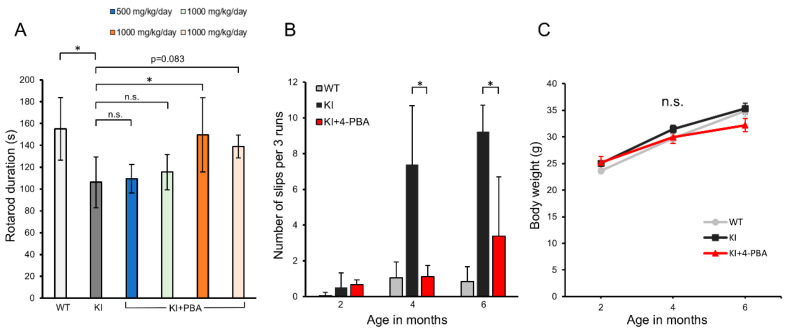
Effects of 4-PBA treatment on motor performance of KI mice. (**A**) Rotarod test results showing latency to fall (in seconds) for WT, KI, and KI+PBA-treated groups at approximately 7 months of age. Data are shown as mean ± SD. Sample sizes (*n*): WT = 23, KI = 9. The KI+PBA groups (blue: *n* = 4, light green: *n* = 3, orange: *n* = 8, light orange: *n* = 6) correspond to the treatment conditions described in Figure 4. (**B**) Number of slips across three trials in WT, KI, and KI+PBA groups across different ages. For the KI+PBA group, mice treated with 1000 mg/kg/day of 4-PBA from 2.5 months of age for 2 months (identified in panel B as the group with the best performance in the rotarod test) were used. Sample sizes (*n*): WT = 16, KI = 13, KI+4-PBA = 6. Data are shown as mean ± SD. (**C**) Changes in body weight over time in WT (*n* = 35), KI (*n* = 31), and KI+PBA (*n* = 6) groups. Data are shown as mean ± SEM. Data are presented as mean ± SD or SEM, as appropriate. Statistical analyses were performed as follows: (**A**) One-way ANOVA revealed a significant group difference (F(5, 47) = 6.08, *p* = 0.0002). Dunnett’s post hoc test: WT vs. KI, *t* = 4.63, *p* < 0.001; KI vs. KI+PBA (blue), *t* = 0.20, *p* = 0.9998; KI vs. KI+PBA (light green), *t* = 0.54, *p* = 0.9791; KI vs. KI+PBA (orange), *t* = 3.45, *p* = 0.0055; KI vs. KI+PBA (light orange), *t* = 2.40, *p* = 0.083. (**B**) The Kruskal–Wallis test followed by Bonferroni-corrected pairwise Mann–Whitney *U* tests showed significantly fewer foot slips in KI+PBA vs. KI at 4 months 4 months (*U* = 72, *p* = 0.0103) and 6 months (*U* = 69.5, *p* = 0.013), indicating improved motor coordination. (**C**) Body weight was analyzed by one-way ANOVA at 2 months (F(2, 65) = 2.853, *p* = 0.0649) and by Kruskal–Wallis at 4 and 6 months (χ^2^(2) = 3.41, *p* = 0.1819; χ^2^(2) = 1.45, *p* = 0.4834), with no significant differences among WT, KI, and KI+PBA groups. Significance: *p* < 0.05 (*); n.s., not significant.

**Figure 2 ijms-26-09779-f002:**
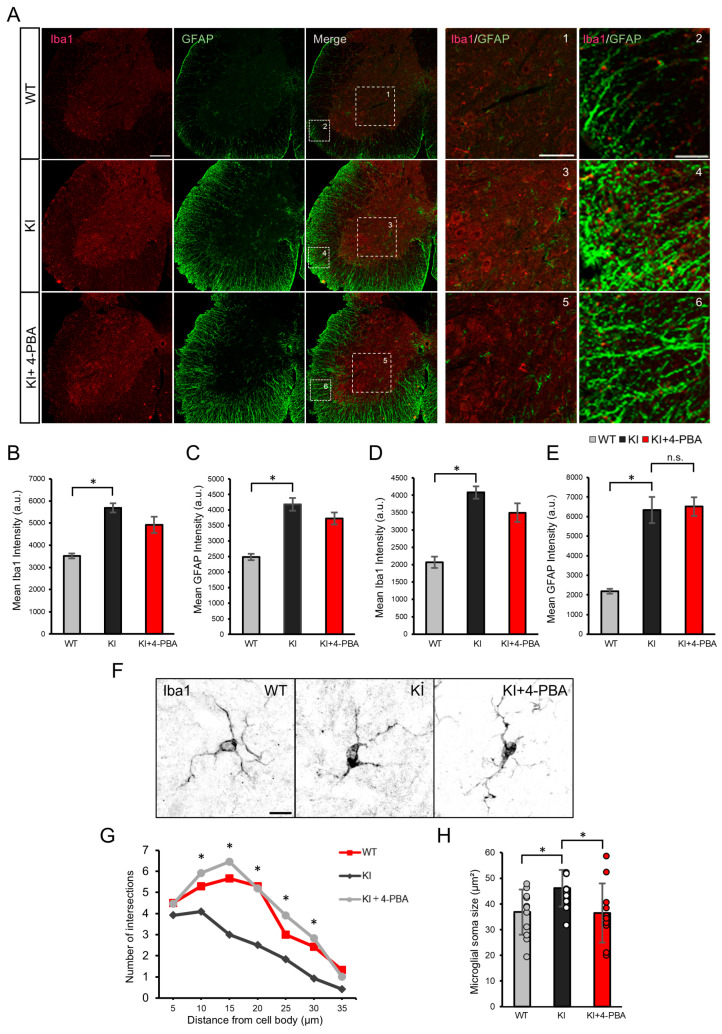
GFAP and Iba1 expression in the spinal cord of *Ubap1* mutant mice and the effect of 4-PBA treatment. (**A**) Representative immunofluorescence images of spinal cord sections from wild-type (WT), *Ubap1* knock-in (KI), and KI mice treated with 4-phenylbutyric acid (KI+4-PBA), stained with antibodies against Iba1 (red) and GFAP (green). Scale bars: 200 µm. Merged images are shown in the third column. White dashed boxes indicate regions shown at higher magnification on the right (boxes 1–6). Scale bars: 100 µm (left), 50 µm (right). (**B**–**E**) Quantification of mean fluorescence intensities of Iba1 and GFAP in the anterior horn (**B**,**C**) and white matter (**D**,**E**) of the spinal cord. Both Iba1 and GFAP expression levels significantly increased in KI mice. The Iba1 expression level clearly increased in KI mice and slightly decreased with 4-PBA treatment (*p* = 0.13). Sample sizes: WT (*N* = 4 mice, *n* = 8 sections), KI (*N* = 4 mice, *n* = 8 sections), KI+4-PBA (*N* = 3 mice, *n* = 6 sections). (**F**) Representative high-magnification images of Iba1-positive microglia in the anterior horn. Scale bar: 10 µm. (**G**) Sholl analysis showing the number of microglial process intersections as a function of distance from the cell body. KI+4-PBA mice showed increased ramification compared with KI mice (*p* < 0.05). Sample sizes: WT (*N* = 3 mice, *n* = 7 cells), KI (*N* = 3 mice, *n* = 12 cells), KI+4-PBA (*N* = 3 mice, *n* = 11 cells). (**H**) Quantification of microglial soma size. Sample sizes: WT (*N* = 4 mice, *n* = 12 cells), KI (*N* = 3 mice, *n* = 14 cells), KI+4-PBA (*N* = 3 mice, *n* = 11 cells). Sholl analysis showing the number of microglial process intersections as a function of distance from the cell body. KI+4-PBA mice showed enhanced ramification compared with KI mice (*p* < 0.05). Data are presented as mean ± SEM. Statistical analyses were performed as follows: (**B**) Kruskal–Wallis test: χ^2^(2) = 13.445, *p* = 0.0012; Bonferroni-corrected Mann–Whitney *U* test: WT vs. KI, *U* = 64, *p* = 0.00046; WT vs. KI+PBA, *U* = 43, *p* = 0.03810; KI vs. KI+PBA, *U* = 35, *p* = 0.54300. (**C**) One-way ANOVA: F(2, 19) = 28.16, *p* = 2.08 × 10^−7^; Bonferroni-corrected *t*-test: WT vs. KI, *p* = 0.0000019; WT vs. KI+PBA, *p* = 0.00028; KI vs. KI+PBA, *p* = 0.24914. (D) One-way ANOVA: F(2, 19) = 29.52, *p* = 1.48 × 10^−6^; Bonferroni-corrected *t*-test: WT vs. KI, *p* = 0.000013; WT vs. KI+PBA, *p* = 0.00028; KI vs. KI+PBA, *p* = 0.17206. (**E**) Kruskal–Wallis test: χ^2^(2) = 14.616, *p* = 0.0006703; Bonferroni-corrected Mann–Whitney *U* test: WT vs. KI, *U* = 64, *p* = 0.00046; WT vs. KI+PBA, *U* = 48, *p* = 0.00200; KI vs. KI+PBA, *U* = 23, *p* = 1.00000. (**G**) Kruskal–Wallis tests were performed at each radial distance from the soma, followed by Bonferroni-corrected Mann–Whitney *U* tests comparing KI and KI+PBA groups. The results were as follows: 10 µm, χ^2^(2) = 6.51, *p* = 0.0386; KI vs. KI+PBA, *U* = 24.5, *p* = 0.03000. 15 µm, χ^2^(2) = 14.66, *p* = 0.000655; KI vs. KI+PBA, *U* = 12, *p* = 0.00250. 20 µm, χ^2^(2) = 12.49, *p* = 0.00195; KI vs. KI+PBA, *U* = 21, *p* = 0.01650. 25 µm, χ^2^(2) = 6.62, *p* = 0.0365; KI vs. KI+PBA, *U* = 28, *p* = 0.05000. 30 µm, χ^2^(2) = 8.94, *p* = 0.0114; KI vs. KI+PBA, *U* = 18, *p* = 0.00760. No significant group differences were detected at 5 µm and 35 µm: 5 µm, χ^2^(2) = 3.33, *p* = 0.1888; 35 µm, χ^2^(2) = 3.34, *p* = 0.1880. (**H**) One-way ANOVA: F(2, 34) = 4.67, *p* = 0.0161; Tukey’s HSD test: KI vs. KI+PBA, *p* = 0.03326; WT vs. KI, *p* = 0.03771; WT vs. KI+PBA, *p* = 0.93207. Significance: *p* < 0.05 (*); n.s., not significant.

**Figure 3 ijms-26-09779-f003:**
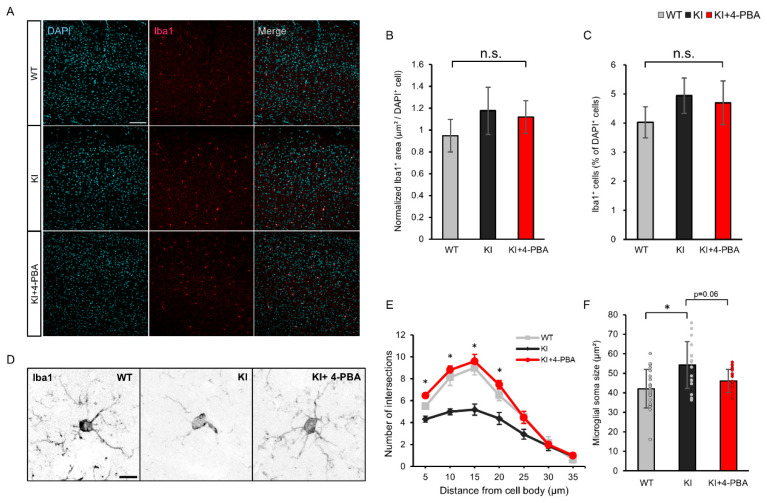
Microglial morphology in the cerebral cortex of *Ubap1* mutant mice and the effect of 4-PBA treatment. (**A**) Representative immunofluorescence images of cerebral cortex sections from wild-type (WT), *Ubap1* knock-in (KI) mice, and KI mice treated with 4-phenylbutyric acid (KI+4-PBA), stained with DAPI (cyan) and Iba1 (red). Merged images are shown. Scale bar: 100 µm. (**B**,**C**) Quantification of Iba1-positive area normalized to DAPI-positive cell number (**B**) and percentage of Iba1-positive cells among DAPI-positive cells (**C**). Sample sizes: WT (*N* = 4 mice, *n* = 10 sections), KI (*N* = 5 mice, *n* = 7 sections), KI+4-PBA (*N* = 4 mice, *n* = 7 sections). (**D**) Representative high-magnification images of Iba1-stained microglia in the cerebral cortex of each group. Scale bar: 10 µm. (**E**) Sholl analysis showing the number of microglial process intersections as a function of distance from the cell body. KI+4-PBA mice exhibited increased ramification compared with KI mice (* *p* < 0.05). Sample sizes: WT (*N* = 4 mice, *n* = 8 cells), KI (*N* = 4 mice, *n* = 16 cells), KI+4-PBA (*N* = 4 mice, *n* = 15 cells). (**F**) Quantification of microglial soma size. KI mice showed enlarged somata, which tended to decrease with 4-PBA treatment (*p* = 0.06). Data are presented as mean ± SEM. Sample sizes: WT (*N* = 4 mice, *n* = 21 cells), KI (*N* = 4 mice, *n* = 17 cells), KI+4-PBA (*N* = 4 mice, *n* = 17 cells). Data are presented as mean ± SEM (**B**,**C**,**E**) or mean ± SD (**F**). Statistical analyses were performed as follows: (**B**) One-way ANOVA: F(2, 21) = 0.357, *p* = 0.704; Tukey’s HSD test: KI vs. KI+PBA, *p* = 0.87921; WT vs. KI, *p* = 0.94560; WT vs. KI+PBA, *p* = 0.67996. (**C**) Kruskal–Wallis test: χ^2^(2) = 2.09, *p* = 0.351; Bonferroni-corrected Mann–Whitney *U* test: WT vs. KI, *U* = 49, *p* = 0.58; WT vs. KI+PBA, *U* = 44, *p* = 1.00; KI vs. KI+PBA, *U* = 29, *p* = 1.00. (**E**) Kruskal–Wallis tests were performed at each radial distance from the soma, followed by Bonferroni-corrected Mann–Whitney *U* tests comparing KI and KI+PBA groups. The results were as follows: 5 µm, χ^2^(2) = 20.045, *p* = 0.0000444; KI vs. KI+PBA, *U* = 18, *p* = 0.00012. 10 µm, χ^2^(2) = 24.405, *p* = 0.00000502; KI vs. KI+PBA, *U* = 4.5, *p* = 0.000013. 15 µm, χ^2^(2) = 21.054, *p* = 0.0000268; KI vs. KI+PBA, *U* = 13.5, *p* = 0.000074. 20 µm, χ^2^(2) = 15.904, *p* = 0.000352; KI vs. KI+PBA, *U* = 29, *p* = 0.00088. No significant group differences were detected beyond 25 µm: 25 µm, χ^2^(2) = 5.202, *p* = 0.0742; 30 µm, χ^2^(2) = 0.0955, *p* = 0.953; 35 µm, χ^2^(2) = 0.882, *p* = 0.6435. (**F**) Kruskal–Wallis test: χ^2^(2) = 10.582, *p* = 0.00504; Steel–Dwass test: WT vs. KI, *U* = 280, *p* = 0.00811; WT vs. KI+PBA, *U* = 225, *p* = 0.35925; KI vs. KI+PBA, *U* = 210.5, *p* = 0.05958. Significance: *p* < 0.05 (*); n.s., not significant.

**Figure 4 ijms-26-09779-f004:**
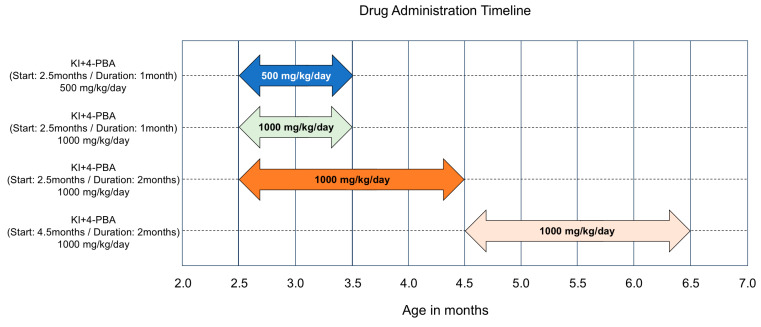
Drug administration timeline. Drug administration timeline. KI mice were treated with 4-PBA under the following conditions: 500 mg/kg/day from 2.5 months of age for 1 month (*n* = 4). 1000 mg/kg/day from 2.5 months of age for 1 month (*n* = 3). 1000 mg/kg/day from 2.5 months of age for 2 months (*n* = 8). 1000 mg/kg/day from 4.5 months of age for 2 months (*n* = 6).

## Data Availability

The data that support the findings of this study are available from the corresponding author upon reasonable request.

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
