# Peer review of "4-Phenylbutyric Acid Improves Gait Ability of UBAP1-Related Spastic Paraplegia Mouse Model: Therapeutic Potential for SPG80"

_ijms, 2025, doi:10.3390/ijms26199779_

Round 1

Reviewer 1 Report

Comments and Suggestions for Authors

I read with great interest the article by Shimozono and colleagues.

The authors investigated the use of 4-phenylbutyric acid to improve gait ability in a SPG80, a mouse model of hereditary spastic paraplegia. This is an interesting and well written article with a substantial amount of data.

HSPs are rare disease, but when all forms are considered together (close to 90 forms), they have a significant impact on the health systems world-wide. Thus, therapeutic interventions are highly needed.

I have some comments for the authors.

1) Are there other HSP forms that can benefit from this treatment? It would be very interesting to, at least as hypothesis, to suggest which forms of HSP might benefit. For instance, the authors mention SPG15, which activates microglia. In this regard, while seminal works have been cited, I recommend updating the references to provide a more current view of the field.

2) While did the authors choose to use an “Accelerated rotarod” test instead of a “Constant speed rotarod” test? In the materials and method section, only rotarod test is mentioned. Could the authors clarify and comment this point?

3) The treatment appears to have a significative effect on motor function in the mouse model. How might this recovery be translated to humans? In other words, are the results sufficient to suggest a therapeutic effects in patients?

4) Please, provide a clearer legend for figure 1, as it is currently not clear. In panel A how may treated animals were used? Please, also explain the rationale for the numbers of WT in comparison to KI animals. What is meant by “the most favourable outcome”?

5) Since it is clear that the treatment did not rescue the microglia inflammation, how do the authors interpret this finding in relation to the apparent improvement in motor capacity?

6) the suthors should comment about future directions, such as testing higher concentration of the drug and addressing whether it could be toxic.

7) Could this drug be used readily in humans?

Author Response

Reply to reviewer 1.

Thank you for your useful and kind suggestions and comments. We revised our manuscript according to your suggestions and comments, and the changes are highlighted in red.

Reviewer 1:

Q1. Are there other HSP forms that can benefit from this treatment? It would be very interesting to, at least as hypothesis, to suggeat which forms of HSP might benefit. For instance, the authors mention SPG15, which activates microglia. In this regard, while seminal works have been cited, I recommend updating the references to provide a more current view of the field.

Response.

We thank the reviewer for the varluable suggestion. We agree that other HSP subtypes may also benefit from interventions targeting proteostasis. As recommended, we expanded the Discussion and updated the references to reflect recent advances in the field. In addition to UBAP1-related HSP, we described that subtypes such as SPG11, SPG15, SPG4, SPG20, and SPG53 can also relevant, as they share features like microglial activation, lysosomal stress, or ESCRT dysfunction (page 13, lines 20-23).

Given that these pathological features converge on impaired proteostasis, chemical chaperones like 4-PBA could offer broader therapeutic potential. However, direct experimental validation in each subtype will be necessary to confirm this possibility. (pages 13, lines 23-24; page 14, lines 1-2).

Q2. While did the authors choose to use an “Accelerated rotarod” test instead of a “Constant speed rotarod” test? In the materials and method section, only rotarod test is mentioned. Could the authors clarify and comment this point?

Response.

We apologize for the lack of clarity. We used an accelerated rotarod protocol (4-40 rpm over 300 s) and have now specified this in the Materials and Methods section (page 16, lines 6-11). We selected this paradigm because it is more sensitive for detecting motor coordination deficits in HSP models. Previous studies employing different HSP mouse models have also used the accerelated rotarod test [References 43 and 44], supporting its suitability for evaluating motor performance in our study (page 16, lines 11-13).

Q3. The treatment appears to have a significant effect on motor function in the mouse model. How might this recovery be translated to humans? In other words, are the results sufficient to suggest a therapeutic effects in patients?

Response.

We thank the reviewer for raising this important issue. Although our results demonstrate significant functional recovery in the Ubap1 knock-in mouse model, direct translation to patients requires caution. We added a statement in the Discussion (the last paragraph) emphasizing that 4-PBA is already approved for clinical use in urea cyclic disorders, which supports the feasibility of reporposing. However, kee issues such as optimal dosing, CNS penetration, and long-term safety in HSP patients must be carefully evaluated in clinical trials. Our study provides preclinical evidence, but not sufficient proof of therapeutic efficacy in patients,

Q4. Please, provide a clearer legend for figure 1, as it is currently not clear. In panel A how many treated animals were used? Please, also explain the rationale for the numbers of WT in comparison to KI animals. What is meant by “favorable outcome”?

Response.

We apologize for the unclear legend in Figure 1. We revised the figure ledgend to improve clarity. The number difference between WT and KI animals was due to animal availability at the time of the experiments, not a specific rationale. Statistical analyses were adjusted accordingly to ensure valid comparisons. We also clarified that “the most favourable outcome” refers to the maximum latency to fall achieved during the rotarod test (page 29, lines 10-11).

Q5. Since it is clear that the treatment did not rescue the microglia inflammation, how do the authors interpret this finding in relation to the apparent improvement in motor capacity?

Response.

We appreciate the reviewer’s insightful comment. We agree that 4-PBA treatment did not normalize the increased number of Iba1-positive microglia, However, our data demonstrate that 4-PBA partially reversed microglial changes, such as hypertrophic soma and reduced branching. This suggests that the treatment attenuated microglial activation. We revised the Discussion (page 12, lines 2-8) to clarify that the improvement in motor function may be associated with modulation of microglial activity, even without a complete rescue of microgliosis.

Q6. The authors should comment about future directions, such as testing higher concentration of the drug and addressing whether it could be toxic.

Response.

We appreciate this suggestion. We added a statement in the Discussion (the last paragraph) about the need to test different concentrations and to evaluate potential toxicity in the future studies.

Q7. Could this drug be used readily in humans?

Response.

We thank the reviewer for raising this important translational aspect. We added a statement in the Discussion (the last paragraphs) emphasizing that 4-PBA is already approved for clinical use in urea cyclic disorders, which supports the feasibility of reporposing.

Reviewer 2 Report

Comments and Suggestions for Authors

The idea, outcome and methodology of the study are quite clear and do not raise major concerns.

Still, some minor issues can be identified.

There is some discrepancy between apparent curative effects of the studied compound (4-PBA) on the biomechanical performance of mutant mice and modest modifications in the glial cells. According to Figure 1, the administration 4-PBA almost completely restores the number of slips and duration of the rotarod test, especially in the mice aged 2 to 4 months. On the other hand, the histochemical part of the study (morphology of microglia and astrocytes) demonstrated only modest, although still detectable, improvements after the 4-PBA application.

In this regard, why were only glial cells studied? Perhaps, the major effect of 4-PBA is located in motoneurons - both upper motoneurones in the motor cortex and lower motoneurones (alpha-motoneurone) in the spinal cord), which actually execute the motor function? I would recommend to discuss this issue in the Discussion section.

Author Response

Reply to reviewer 2.

Thank you for your useful and kind suggestions and comments. We revised our manuscript according to your suggestions and comments, and the changes are highlighted in red.

Reviewer 2:

The idea, outcome and methodology of the study are quite clear and do not raise major concerns.

Still, some minor issues can be identified.

There is some discrepancy between apparent curative effects of the studied compound (4-PBA) on the biomechanical performance of mutant mice and modest modifications in the glial cells. 

According to Figure 1, the administration 4-PBA almost completely restores the number of slips and duration of the rotarod test, especially in the mice aged 2 to 4 months. On the other hand, the histochemical part of the study (morphology of microglia and astrocytes) demonstrated only modest, although still detectable, improvements after the 4-PBA application.

In this regard, why were only glial cells studied? Perhaps, the major effect of 4-PBA is located in motoneurons – both uppermotoneurons in the motor cortex and lower motoneurons (alpha-motoneuron) in the spinal cord, which actually execute the motor function? I would recommend to discuss this issue in the Discussion section.

Response.

We appreciate the reviewer’s insightful comments. As suggested, we addressed this issue in the Discussion section. In our study, we focused on glial responses because microglial and astroglial activation are key features of neuroinflammation and are increasingly recognized as important contributors to the pathogenesis of hereditary spastic paraplegia. However, we agree with the reviewer that motoneurons, including cortical upper motoneurons and spinal alpha motoneurons, play a central role in motor function and are likely to be major targets of disease pathology.

 We prioritized glial markers in this study due to their relevance in neuroinflammation and the availability of well-established histological tools. Nonetheless, we acknowledge that analyzing motoneuronal pathology would provide complementary insights. The discrepancy between the robust behavioral improvements and the modest glial modulation suggests that 4-PBA may exert additional protective effects on motoneurons.

 Although direct evidence for 4-PBA-mediated neuroprotection in motoneurons is currently limited, its known effects on reducing endoplasmic reticulum stress and improving protein homeostasis suggest a plausible mechanism by which it could support motoneuronal function. This may help explain the behavioral improvements observed in our study. Further investigations are needed to directly assess motoneuronal integrity and function following 4-PBA treatment.

 To address this point, we added a paragraph in the Discussion section (page 12, lines 11-21). In that paragraph, we discuss the possibility that 4-PBA may have neuroprotective effects on motoneurons and propose future studies to explore this hypothesis. These studies may include immunohistochemical analysis of motoneuronal markers such as choline acetyltransferase (ChAT) and NeuN, as well as electrophysiological evaluations to better understand the therapeutic mechanisms of 4-PBA.

Round 2

Reviewer 1 Report

Comments and Suggestions for Authors

Dear Authors,

you addressed all the comments made by this Reviewer.

Congratulation for the interesting paper.

Author Response

Thank you for your useful suggestions and comments.